# Enhancing Spatial Variability Representation of Radar Nowcasting with Generative Adversarial Networks

Aofan Gong [1], Ruidong Li [1], Baoxiang Pan [2], Haonan Chen [3], Guangheng Ni [1,*] and Mingxuan Chen [4]

[1] Department of Hydraulic Engineering, Tsinghua University, Beijing 100084, China; gaf20@mails.tsinghua.edu.cn (A.G.); lrd19@mails.tsinghua.edu.cn (R.L.)
[2] Institute of Atmospheric Physics, Chinese Academy of Sciences, Beijing 100029, China; panbaoxiang@lasg.iap.ac.cn
[3] Electrical and Computer Engineering, Colorado State University, Fort Collins, CO 80523, USA; haonan.chen@colostate.edu
[4] Institute of Urban Meteorology, China Meteorological Administration, Beijing 100089, China; mxchen@ium.cn
[*] Correspondence: ghni@tsinghua.edu.cn

**Abstract:** Weather radar plays an important role in accurate weather monitoring and modern weather forecasting, as it can provide timely and refined weather forecasts for the public and for decision makers. Deep learning has been applied in radar nowcasting tasks and has exhibited a better performance than traditional radar echo extrapolation methods. However, current deep learning-based radar nowcasting models are found to suffer from a spatial "blurry" effect that can be attributed to a deficiency in spatial variability representation. This study proposes a Spatial Variability Representation Enhancement (SVRE) loss function and an effective nowcasting model, named the Attentional Generative Adversarial Network (AGAN), to alleviate this blurry effect by enhancing the spatial variability representation of radar nowcasting. An ablation experiment and a comparison experiment were implemented to assess the effect of the generative adversarial (GA) training strategy and the SVRE loss, as well as to compare the performance of the AGAN and SVRE loss function with the current advanced radar nowcasting models. The performances of the models were validated on the whole test set and inspected in two storm cases. The results showed that both the GA strategy and SVRE loss function could alleviate the blurry effect by enhancing the spatial variability representation, which helps the AGAN to achieve better nowcasting performance than the other competitor models. Our study provides a feasible solution for high-precision radar nowcasting applications.

**Keywords:** nowcasting; radar; generative adversarial network; spatial variability

## 1. Introduction

Detailed weather forecasting over a very short period that lasts from the present to the next few hours, which is also known as nowcasting, has significant benefits related to weather-related human activities, including public traffic, flood alarms, disaster warnings, emergency management, and risk prevention [1]. Based on current meteorological observations, accurate nowcasting can provide timely (up to the minute level) and refined (mesoscale or even microscale) weather forecasts for the public and decision makers [2].

Thanks to the rapid progress in meteorological observation technology, Doppler weather radars have become one of the most valuable tools for observing clouds, precipitation, and wind [3,4]. Since radars can detect larger areas than rain gauges and scan at a higher resolution in shorter intervals than satellites, they can better reflect the spatial and temporal variability of the above meteorological elements than rain gauges and satellites, which are also powerful tools for weather forecasting [5–8]. Traditional radar echo extrapolation methods are widely utilized as the basis of nowcasting systems, such as the following: Thunderstorm Identification, Tracking, And Nowcasting (TITAN) [9]; Storm Cell Identification and Tracking (SCIT) [10]; Tracking Reflectivity Echoes by Correlation

(TREC) [11]; the McGill Algorithm for Precipitation Nowcasting by Lagrangian Extrapolation (MAPLE) [12]; Dynamic and Adaptive Radar Tracking of Storms (DARTS) [13]; optical flow-based methods [14,15]; and methods for nowcasting the growth and decay of storms [16]. These traditional methods have exposed shortcomings due to the limits of their underlying assumptions and constraints, including the motion–field constancy of TREC-based methods and the spatial smoothness constraints of optical flow-based methods [17–19].

In recent years, deep learning (DL), which has seen remarkable advancements in diverse domains, such as computer vision [20], natural language processing [21], and geoscience [22], has also been applied in radar nowcasting by meteorological researchers [23]. In these studies, radar nowcasting was formulated as a spatiotemporal sequence extrapolation problem. Compared to conventional extrapolation models, DL models usually perform better because of their strong non-linear modeling capacity driven by large-scale historical radar echo datasets [24]. Generally, DL nowcasting models consist of two main types: the convolutional neural network (CNN) and the recurrent neural network (RNN). CNNs are widely adopted in image processing because of their translation invariance property, while RNNs feature in time series analysis because of their recurrent structure. Present CNN-based nowcasting models focus more on the spatial correlation of meteorological fields, while RNNs pay more attention to the sequential correlation [25]. For CNNs, several researchers have made progress in developing three-dimensional CNNs and their variants [26–28]. Klein, et al. [29] proposed a dynamic convolutional layer, but revealed the limits of predicting one echo frame in one step. Ayzel, et al. [30] designed an All Convolutional Neural Network and introduced a more effective model called "Rain-Net" with the U-Net [31] structure in their later work [32]. Trebing, et al. [33] designed the SmaAt-UNet with attentional modules and depthwise separable convolutions, which produced higher performance with fewer parameters than the original U-Net. For RNNs, the authors in [34] proposed the Convolutional Long Short-Term Memory (ConvLSTM), which replaced the full connection in the gates of the vanilla LSTM [35] with a convolutional operator. This work is regarded as the pioneer study of DL-based precipitation nowcasting. An encoding–forecasting network structure was built, based on the same authors' newly proposed Trajectory Gated Recurrent Unit (TrajGRU) in their following study [36]. Wang, et al. [37] proposed the PredRNN by expanding the original ConvLSTM with spatiotemporal memory flow and developed an enhanced model, "PredRNN++", in their following study [38]. Wu, et al. [39] proposed the MotionRNN, which significantly improves the ability to predict changeable motions and avoid motion vanishing for stacked multiple-layer nowcasting models.

Although DL models have shown advantages in radar nowcasting tasks, several researchers have noted a systematic deviation from current DL models and summarized them as "blurry" effects [32,36,40,41]. DL models were found to neglect high-intensity features and small-scale patterns of the weather system, causing the generated images to lose spatial variability and look blurry. This effect was attributed to the impact of convolutional operators contained in DL models, in that their inductive bias of the translation invariance would lead to a loss of spatiotemporal features for precipitation [42,43]. Under this deviation, although DL models can outperform traditional models in regard to most precision scores, they are weak when it comes to learning the spatial variability of the radar echo sequences. Since radar parameters have a strong relationship with precipitation, the deviation in the spatial variability representation of radar echoes causes an expanded error in the downstream applications, such as in urban flood simulation [44,45].

Researchers have adopted generative adversarial networks (GANs) [46] to alleviate the blurry effect. Jing, et al. [40] developed an Adversarial Extrapolation Neural Network (AENN), based on the ConvLSTM and CNN, for nowcasting at an interval of 30 min to generate accurate and realistic extrapolation echoes. Tian, et al. [41] proposed a Generative Adversarial Convolutional Gated Recurrent Unit (GA-ConvGRU) model that outperformed the original ConvGRU, but their model suffered due to training instability.



Xie, et al. [47] developed a more robust Energy-Based Generative Adversarial Forecaster and demonstrated the stability and accuracy of their model. Ravuri, et al. [48] compared their novel Deep Generative Model of Rainfall (DGMR) with ensemble optical flow and evaluated its effectiveness with both quantitative verification measures and qualitative cognitive assessments. However, few studies have paid attention to enhancing the local spatial variability representation ability of DL nowcasting models.

This study aimed to solve the above drawbacks of DL models for radar nowcasting. A Spatial Variability Representation Enhancement (SVRE) loss function and an Attentional Generative Adversarial Network (AGAN) are proposed to enhance the spatial variability representation of radar nowcasting. The SVRE loss implements regularization in the adversarial training process with representative spatial variability features. The SVRE loss function and the AGAN model are evaluated on a three-year radar observation dataset through an ablation experiment and a comparison experiment. Several state-of-the-art DL nowcasting models are selected as the baseline models, including the MotionRNN [49], the SmaAt-UNet [33], and a traditional ensemble nowcasting model, PySTEPS [50]. The rest of this article is organized as follows. Section 2 illustrates the principle of the SVRE loss function and the architecture of the AGAN. Details of the experiments are explained in Section 3, and the results of these experiments are presented and discussed in Section 4. The last section concludes this study and points out the direction of future work.

## 2. Methods

### 2.1. Problem Statement

As previous studies [34] summarized, the radar nowcasting problem can be abstracted as a spatiotemporal sequence extrapolation problem which aims to predict the next length-$n$ sequence given a previous length-$m$ observation. Let tensor $\mathcal{X} \in \mathbb{R}^{N \times H \times W}$ denote radar observations over a previous period (with length $N$, height $H$, and width $W$) and $\theta$ denote the parameters of a DL-based nowcasting model; then, the problem can be described by

$$\hat{\mathcal{X}}_{1:n} = \underset{\mathcal{X}_{1:n}}{\mathrm{argmax}}\, p(\mathcal{X}_{1:n}|\mathcal{X}_{1-m:0};\theta), \tag{1}$$

where the subscript index of $\mathcal{X}$ denotes the tensor's slice at the corresponding time step (0 represents the current time).

The perspective of generative models in machine learning can change to a probabilistic problem instead of a deterministic problem, which means that the prediction is no longer estimated by the maximum likelihood estimation of the conditional probabilistic distribution, but is, rather, sampled from the conditional probabilistic distribution given the prior distribution of a latent code $\mathcal{Z}$, used to describe the latent states of the system, and described as

$$
\begin{aligned}
\hat{\mathcal{X}}_{1:n} &\sim p(\mathcal{X}_{1:n}|\mathcal{X}_{1-m:0};\theta) \\
&= \int_{\mathcal{Z}} p(\mathcal{X}_{1:n}, \mathcal{Z}|\mathcal{X}_{1-m:0};\theta)\mathrm{d}\mathcal{Z} \\
&= \int_{\mathcal{Z}} p(\mathcal{X}_{1:n}|\mathcal{X}_{1-m:0}, \mathcal{Z};\theta)p(\mathcal{Z}|\mathcal{X}_{1-m:0})\mathrm{d}\mathcal{Z} \\
&= \mathbb{E}_{\mathcal{Z}}[p(\mathcal{X}_{1:n}|\mathcal{X}_{1-m:0}, \mathcal{Z};\theta)].
\end{aligned}
\tag{2}
$$

### 2.2. The Principle of GAN

Machine learning models generally consist of two paradigms: discriminative models and generative models. Fundamentally, discriminative models aim to draw the decision boundaries from the data space, while generative models learn a joint probability pattern based on Bayesian rules, thus learning and applying the mapping of the low-dimensional manifold to the high-dimensional data space [51,52]. Since generative models calculate the joint distribution of the input and the target variables before the derivation of the posterior distribution, they can learn more information and, thus, describe more indicative features of the data. A generative model can extract more information about the relationship between

the input and the target variables than a discriminative model can, especially when latent variables exist. Classic generative models, including Gaussian mixture models, hidden Markov models, Boltzmann machines, and variational autoencoders, calculate the joint distribution by explicitly specifying the probabilistic density function and optimizing it with suitable optimization algorithms, such as gradient descent or variational inference [51].

The GAN was one of the generative models proposed by Goodfellow, et al. [53], which was designed to identify the joint distribution based on the adversarial learning theory. A GAN is composed of a generator that captures the data distribution and a discriminator that estimates the probability of where a sample came from. During adversarial learning, the generator is optimized with the guidance of the discriminator. The optimization goal of a GAN is to achieve a Nash equilibrium between the generator and the discriminator. The optimization of a GAN is equivalent to a min–max two-player game between the generator and the discriminator, expressed by

$$\min_{G} \max_{D} V(D, G) = \mathbb{E}_{\mathbf{x} \sim p_{data}(\mathbf{x})}[\log D(\mathbf{x})] + \mathbb{E}_{\mathbf{z} \sim p_{\mathbf{z}}(\mathbf{z})}[\log(1 - D(G(\mathbf{z})))], \qquad (3)$$

where $D$ and $G$ represent discriminator and generator, respectively. $\mathbf{x}$ represents the sample and $\mathbf{z}$ denotes the latent random vector of the generator. The authors proved that the generator and the discriminator reach Nash equilibrium if, and only if, the estimated distribution equals the data distribution by simultaneously training the generator and the discriminator, given there are enough data (in practice, they have to be trained alternately instead). As a result of the outstanding performance of the GAN, this generative model has been applied as one of the mainstream tools for generative learning.

*2.3. The SVRE Loss Function*

As its name suggests, the SVRE loss function focuses on the direct enhancement of a model's spatial variability representation. The optimization of the original GAN proved to be equivalent to a sigmoid cross-entropy function [53], while Mao, et al. [54] pointed out that this function had a gradient vanishing problem, leading to quality loss in the generated images. Therefore, we adopted the least-squares loss (proposed by the above authors) as the basic adversarial loss. Beyond generative adversarial training, we added two additional regularization terms to the adversarial loss to improve the sharpness and spatial variability of the generated images. The first regularization term is L1-normalization, which has been used for encouraging less blurring in image translation tasks [55]. The second term is the L1-norm distance between the coefficient of variation ($C_v$) of the prediction and the index of the target sequence, which has never been used for regularization in previous related studies. In statistics, $C_v$ is defined as the quotient of the standard deviation divided by the mean value of a group of samples. This metric has also been used in hydrological research to describe the precipitation variability and to compare the variability of different precipitation fields [56]. In our study, the standard deviation and the mean value were calculated along the spatial axis ($H$ and $W$), instead of the temporal axis ($N$), to represent spatial variability, as is shown in Equation (4). This L1-norm term of $C_v$ gives a quantification of the gap between the prediction's spatial variability and the observation's spatial variability. Both regularization terms were scaled by a corresponding hyperparameter $\lambda$. The loss functions for the adversarial training of the discriminator and the generator are illustrated in Equations (5) and (6).

$$C_v(\mathcal{X}_t) = \frac{\sigma_{h,w}(\mathcal{X}_t)}{\text{mean}_{h,w}(\mathcal{X}_t)} \qquad (4)$$

$$\mathcal{L}_D(\phi) = \mathbb{E}_{\mathcal{X}}[D(\mathcal{X}_{1:n}|\mathcal{X}_{1-m:0}; \phi) - 1]^2 + \mathbb{E}_{\mathcal{X},\mathcal{Z}}\left[D(G(\mathcal{Z}|\mathcal{X}_{1-m:0}; \theta)|\mathcal{X}_{1-m:0}; \phi)^2\right] \qquad (5)$$

$$\mathcal{L}_G(\theta) = \mathbb{E}_{\mathcal{X},\mathcal{Z}}[D(G(\mathcal{Z}|\mathcal{X}_{1-m:0};\theta)|\mathcal{X}_{1-m:0};\phi) - 1]^2 + \lambda_r\mathbb{E}_{\mathcal{X},\mathcal{Z}}\|G(\mathcal{Z}|\mathcal{X}_{1-m:0};\theta) - \mathcal{X}_{1:n}\|_1$$
$$+ \lambda_v\mathbb{E}_{\mathcal{X},\mathcal{Z}}\|C_v(G(\mathcal{Z}|\mathcal{X}_{1-m:0};\theta)) - C_v(\mathcal{X}_{1:n})\|_1 \tag{6}$$

Here, $D$ and $G$ represent discriminator and generator, respectively. The operator mean denotes the mean value across axis $h$ and $w$ of an image at time $t$ and $\sigma$ denotes its standard deviation. The discriminator loss is meant to minimize the sum of the averaged squared distance between a sample and its corresponding label (0 for fake and 1 for real) to distinguish the predicted sequence from the target. Meanwhile, the generator loss is meant to minimize the averaged squared distance between the fake sample and the real label, since it ultimately needs to confuse the discriminator.

### 2.4. The Architecture of the AGAN

We now present the details of the AGAN (Figure 1). Like other GANs, it consists of a generator and a discriminator. The generator (Figure 1a) follows a U-Net structure, which has been effective in related nowcasting applications. It accepts a 1-h (10 steps with a 6 min interval) historical radar observation sequence (orange rectangles) as the input and predicts the radar sequence for the next hour. The encoding part of the U-Net contains four downscaling blocks (red arrows), which are preceded by a 1 × 1 convolution (orange arrows) for temporal feature combination. The decoding part applies the same structure, containing four upscaling blocks and an additional 1 × 1 convolution. The multi-scale encoding feature maps (blue rectangles) are copied through skip connections (gray arrows) and concatenated with the corresponding decoding feature maps (yellow rectangles) in upscaling blocks (green arrows). The discriminator (Figure 1b) is a fully convolutional network composed of two convolutional blocks and six downscaling blocks, which accepts the concatenated sequences from historical and future data (orange rectangles) as input and returns a probability between 0 and 1 as output to guide the optimization of the generator through adversarial training.

The downscaling and the upscaling blocks of the AGAN are displayed in Figure 2a,b, respectively. For the downscaling blocks, the input feature maps are first reshaped by a 2 × 2 max-pooling layer (the red rectangle) and then fed into two 3 × 3 convolutional layers, each followed by a batch normalization layer (BN) and a rectified linear unit (ReLU) layer. For the upscaling blocks, the input features are reshaped by a 2 × 2 bilinear interpolation layer (the green rectangle) and concatenated with the corresponding encoding feature map copied from the encoder. A convolutional block attention module (CBAM) [57], which was developed for self-adaptive feature combination, is integrated at the end of the scaling block (the orange rectangle) to strengthen the model's attention-based feature refinement ability. This differentiates our model from the original U-Net, and that is why it is called "attentional".

The number of convolutional kernels doubles after downscaling and is reduced by half after upscaling in the generator (32, 64, 128, and 256 kernels in the corresponding blocks), which references the original U-Net. The AGAN contains, in total, about 6.40 million trainable parameters, among which 3.66 million are for the generator and 2.74 million are for the discriminator.

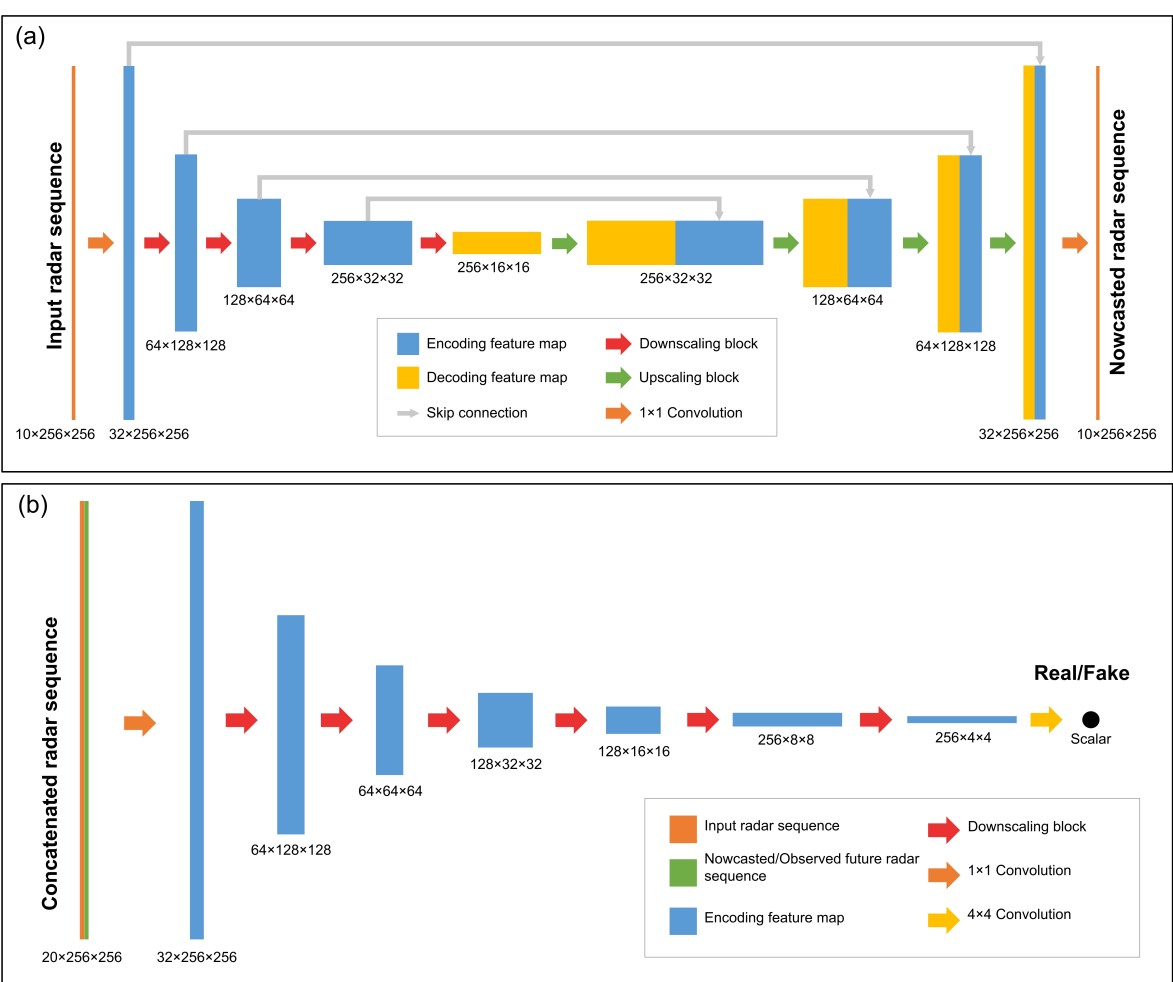

**Figure 1.** Architecture of AGAN. (**a**) AGAN's generator. (**b**) AGAN's discriminator.

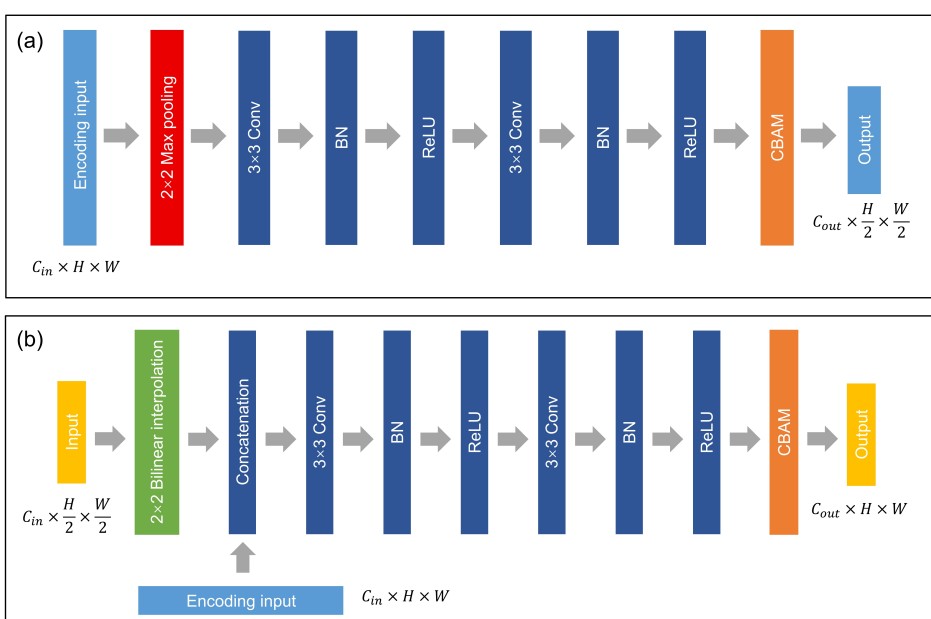

**Figure 2.** Scaling blocks in AGAN. (**a**) Downscaling blocks. (**b**) Upscaling blocks.

## 3. Experiments

### 3.1. Data and Study Area

To validate the effectiveness of the AGAN, we selected a radar composite reflectivity dataset from the Beijing Auto-Nowcast (BJ-ANC) System that was developed by the Institute of Urban Meteorology, China Meteorological Administration, Beijing [58]. The system collects observations from S-band and C-band Doppler weather radars that are part of the China Next Generation Weather Radar (CINRAD) network and produces radar mosaic images with quality control and mosaic generation algorithms. The radar composite reflectivity mosaic product covers a total area of $800 \times 800$ km$^2$, with a spatial resolution of 1 km and a temporal resolution of 6 min. Since we were concerned about the weather in Beijing and its surroundings, we defined the study area as a square region of $256 \times 256$ km$^2$ that had the Yizhuang Radar (one of the CINRAD radars) located at the exact center (39.81°N, 116.47°E) of the square (Yizhuang is near the center of Beijing City), as shown in Figure 3.

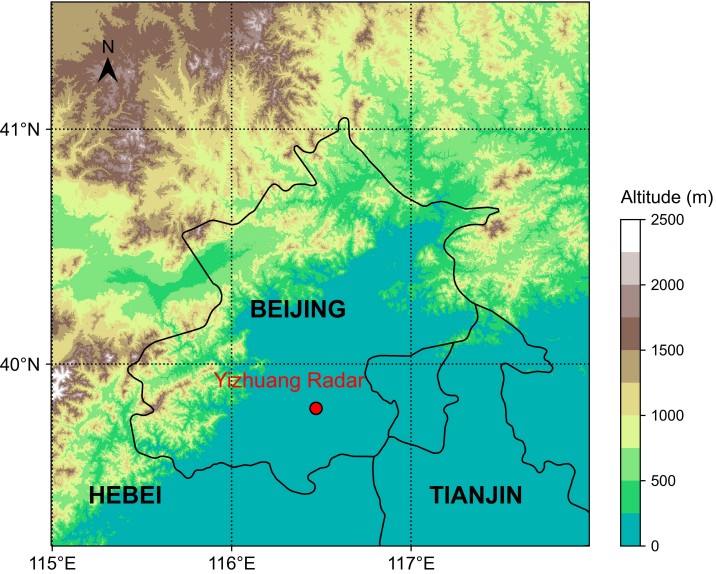

**Figure 3.** The study area (the whole square) and the location of Yizhuang Radar (red circle). The figure is plotted using UTM 50N coordination.

For dataset construction and processing, observations from three S-band radars on 86 rainy days that occurred during the warm seasons (from June to September) from 2017 to 2019 were selected, according to ground precipitation observations. The mosaic images were clipped between 0 and 70 dBZ and scaled with a min–max normalization which converted the values to a range from 0 to 1. According to the input and output settings, we packed observations that covered 20 consecutive time steps into 1 sequence and extracted a total of 18,892 sequences from the dataset. The sequences were sorted in chronological order and temporal overlapping sequences were dropped. Then, they were split into the training set, validation set, and test set with a ratio of approximately 7:1:2. To avoid data leakage, the sequences in the training and validation sets were only selected from the warm seasons in 2017 and 2018, yet all the sequences in the test set came from the warm season in 2019, as shown in Table 1.

**Table 1.** Split settings of the dataset.

| Items | Training | Validation | Test | Removed | Total |
|---|---|---|---|---|---|
| Sequences | 13,200 | 1888 | 3760 | 44 | 18,892 |
| Rainy days | 60 | 9 | 17 | / | 86 |
| Proportion | 0.699 | 0.1 | 0.199 | 0.003 | 1 |
| Year | 2017, 2018 | 2018 | 2019 | / | 2017–2019 |

*3.2. Evaluation Metrics*

The nowcasting performances of different models were evaluated from two aspects. The first one was forecasting accuracy, which has been focused on by most previous studies. The commonly used weather forecast metrics were adopted, including the probability of detection (POD), the false alarm ratio (FAR), and the critical success index (CSI) [59]. These metrics were calculated with a contingency table to count the frequency of hits (*h*), false alarms (*f*), and misses (*m*), as defined in Equations (7)–(9). For a certain threshold, a hit occurs when both the prediction and the target value exceed the threshold within the same grid. A false alarm occurs when the threshold is beyond the target value but under the prediction value, and a miss occurs in the opposite case. In this study, the threshold of these contingency metrics was set to 30 dBZ because this has been commonly used to distinguish heavy rainfall from light rainfall in related studies [60,61]. The mean bias error (MBE), the mean absolute error (MAE), and the root mean squared error (RMSE) were used to roughly estimate the forecasting bias, as defined in Equations (10)–(12).

$$\text{POD} = \frac{h}{h + m} \tag{7}$$

$$\text{FAR} = \frac{f}{h + f} \tag{8}$$

$$\text{CSI} = \frac{h}{h + m + f} \tag{9}$$

$$\text{MBE}(\hat{\mathcal{X}}_t, \mathcal{X}_t) = \text{mean}_{h,w}(\hat{\mathcal{X}}_t - \mathcal{X}_t) \tag{10}$$

$$\text{MAE}(\hat{\mathcal{X}}_t, \mathcal{X}_t) = \text{mean}_{h,w}|\hat{\mathcal{X}}_t - \mathcal{X}_t| \tag{11}$$

$$\text{RMSE}(\hat{\mathcal{X}}_t, \mathcal{X}_t) = \sqrt{\text{mean}_{h,w}(\hat{\mathcal{X}}_t - \mathcal{X}_t)^2} \tag{12}$$

The second aspect was spatial variability representation. Three metrics were adopted to evaluate the spatial variability similarity between the observed precipitation field and the prediction of a certain model: (1) The Jensen–Shannon divergence (JSD), which measures the statistical difference between one probability distribution *p* and a second reference probability distribution $\hat{p}$. It can be proved that the JSD is symmetric to *p* and $\hat{p}$, and ranges from 0 to 1; (2) The structural similarity index, which measures (SSIM) [62] and is used to measure the similarity between two images from the three aspects of luminance, contrast, and structure; (3) The power spectral density (PSD), which presents the relationship between the power and frequency of a signal. It has been used in radar nowcasting tasks to evaluate a model's ability to represent diverse-scale weather patterns [32,48]. In this study, the PSD was calculated both with the height axis and the width axis of the radar images, defined in Equations (13) and (14). The other two metrics are defined in Equations (15) and (16).

$$\text{PSD}_h(\mathcal{X}_t) = 10 \log_{10} \text{mean}_w |\mathcal{F}_h(\mathcal{X}_t)|^2 \tag{13}$$

$$\text{PSD}_w(\mathcal{X}_t) = 10 \log_{10} \text{mean}_h |\mathcal{F}_w(\mathcal{X}_t)|^2 \tag{14}$$

$$\text{JSD}(p_t \| \hat{p}_t) = \frac{1}{2} \sum_{h,w} p_t \log \frac{2p_t}{p_t + \hat{p}_t} + \frac{1}{2} \sum_{h,w} \hat{p}_t \log \frac{2\hat{p}_t}{p_t + \hat{p}_t} \tag{15}$$

$$\text{SSIM}(\hat{\mathcal{X}}_t, \mathcal{X}_t) = \frac{\left[2\text{mean}_{h,w}(\hat{\mathcal{X}}_t) \cdot \text{mean}_{h,w}(\mathcal{X}_t) + C_1\right]\left[2\text{cov}_{h,w}(\hat{\mathcal{X}}_t, \mathcal{X}_t) + C_2\right]}{\left[\text{mean}_{h,w}(\hat{\mathcal{X}}_t)^2 + \text{mean}_{h,w}(\mathcal{X}_t)^2 + C_1\right]\left[\text{std}_{h,w}(\hat{\mathcal{X}}_t)^2 + \text{std}_{h,w}(\mathcal{X}_t)^2 + C_2\right]} \tag{16}$$

In the above equations, $\mathcal{X}_t$ represents the observation radar map at time $t$ and $\hat{\mathcal{X}}_t$ represents the prediction, $p_t$ and $\hat{p}_t$ are the probability distributions of $\mathcal{X}_t$ and $\hat{\mathcal{X}}_t$. std represents the standard deviation operator. The value cov is the covariance between the observation and the prediction. The constants $C_1$ and $C_2$ were set to $1 \times 10^{-4}$ and $9 \times 10^{-4}$ in this research, which was the same as in [62]). Then, $\mathcal{F}_h(\cdot)$ and $\mathcal{F}_w(\cdot)$ denote the Fourier transform operation along the height and width axis, respectively.

### 3.3. Experiment Settings

The performances of the AGAN and the SVRE loss function were evaluated through an ablation experiment and a comparison experiment. In the ablation experiment, the two components, the GA strategy and SVRE, were tested by training the AGAN or the AGAN's generator with or without SVRE loss, respectively. The generator of the AGAN trained with the purely L1-norm loss function served as the control group, denoted by AGAN(g). The names of the models and their meanings are explained in Table 2.

**Table 2.** The names of the models and their meanings in the ablation experiment. The ✓ represents "with" and the × represents "without".

| Model | Trained with GA Strategy | Trained with SVRE Loss |
|---|---|---|
| AGAN(g) | × | × |
| AGAN(g) + SVRE | × | ✓ |
| AGAN | ✓ | × |
| AGAN + SVRE | ✓ | ✓ |

For the AGAN, the generator and the discriminator were trained alternately with an Adam optimizer regularized by a decoupled weight decay of 0.01, where $\beta_1 = 0.9$ and $\beta_2 = 0.999$. The maximum training step was set to 100,000. The Two Time-Scale Update Rule (TTUR) [63] was adopted as one of the adversarial training strategies for better convergence. However, in a way that differed from the original TTUR skill, the learning rate was set to $1 \times 10^{-4}$ for the generator and $5 \times 10^{-5}$ for the discriminator to avoid the early convergence of the discriminator, based on preliminary experiments. The coefficients of the two regularization terms in the loss function were set to 10 and 1 (10 for the reconstruction term and 1 for the SVRE term) according to the performance on the validation set. The early stopping strategy was applied to prevent the model from overfitting. The integration over latent variables in Equation (2) was approximated with six latent random vectors at one training step, which was the same as in another related study [48]. Other settings included a batch size of 8 and the early stopping patience of 10 epochs. The AGAN(g) shared the same settings as the AGAN.

In the comparison experiment, the AGAN with SVRE were compared with three baseline models: the RNN-based MotionRNN [39], the CNN-based SmaAt-UNet [33], and the optical flow-based ensemble forecast system PySTEPS [50]. We kept the architecures of the deep learning baseline models (SmaAt-UNet and MotionRNN) unchanged and implemented the same training strategies on them as on the AGAN. For PySTEPS, the Lucas–Kanade motion tracking method and the semi-Lagrangian extrapolation method were selected.

All the experiments in this study were implemented on a computing platform with an Intel Xeon Gold 6226R CPU and an Nvidia Tesla A100 GPU, based on the open-source machine learning framework PyTorch (https://pytorch.org/ (accessed on 6 August 2022)).

## 4. Results and Discussion

### 4.1. Overall Performance

The overall performances of the models in the ablation experiment and the comparison experiment were evaluated on the whole test set. For each sample in the test set, we first extracted the last frames of the forecast sequence and the observed sequence. Since the time step of the last frame was 60 min ahead of the reference time, we determined the last

frame to have a lead time of 60 min. The evaluation metrics of the observation and the prediction at the lead time of 60 min (+60 min observation and prediction) were calculated to reflect the models' performances for the sample. These metrics were averaged over all samples in the test set. The results of the experiments are listed in Table 3.

**Table 3.** Overall, +60 min nowcasting performances on the test set. The up and down arrows in the heading indicate whether the highest or the lowest was the best for different metrics. A bold number indicates that the model in its row had the best performance, evaluated with the metric in its column.

| Model | POD↑ | FAR↓ | CSI↑ | MBE↓ | MAE↓ | RMSE↓ | SSIM↑ | JSD↓ |
|---|---|---|---|---|---|---|---|---|
| PySTEPS | 0.299 | **0.470** | 0.204 | **0.0** | **6.1** | **9.6** | 0.292 | 0.601 |
| SmaAt-UNet | 0.631 | 0.521 | 0.351 | 13.8 | 14.6 | 18.8 | 0.365 | 0.516 |
| MotionRNN | 0.572 | 0.477 | 0.310 | 14.0 | 15.0 | 19.2 | 0.337 | 0.521 |
| AGAN(g) | **0.749** | 0.568 | 0.373 | 13.7 | 14.6 | 18.8 | 0.360 | 0.437 |
| AGAN(g) + SVRE | 0.643 | 0.484 | 0.377 | 13.2 | 14.4 | 18.6 | 0.377 | 0.477 |
| AGAN | 0.721 | 0.565 | 0.374 | 13.4 | 14.4 | 18.5 | 0.376 | 0.427 |
| AGAN + SVRE | 0.745 | 0.578 | **0.380** | 13.4 | 14.5 | 18.6 | **0.387** | **0.421** |

In Table 3, three baseline models (PySTEPS, SmaAt-UNet, and MotionRNN), the AGAN's generator trained with the ordinary adversarial loss function (AGAN(g)), the AGAN's generator trained with the SVRE loss function (AGAN(g) + SVRE), the AGAN trained with the ordinary adversarial loss function (AGAN), and the AGAN trained with the SVRE loss function (AGAN + SVRE) are included. For forecasting accuracy, it was found that, although the AGAN, AGAN(g)+SVRE, and AGAN + SVRE did not reach the highest POD or the lowest FAR, when considering hits and false alarms together, both the GA strategy and the SVRE loss function could increase the CSI. Compared with the AGAN(g), the AGAN + SVRE increased the CSI from 0.373 to 0.380, indicating that the combination of the GA strategy and SVRE could improve the general forecasting accuracy. It was also observed that SVRE slightly narrowed the MBE, MAE, and RMSE between the prediction and observation. For spatial variability representation, SVRE improved the performances of both the AGAN and AGAN(g) by concurrently increasing the image similarity (increasing the SSIM by 0.073 and 0.050) and reducing the distribution difference (reducing the JSD by 0.085 and 0.100) between the observation and the prediction. The effect of the GA strategy was less significant than for SVRE. Among all the models, the AGAN + SVRE reached the highest CSI, the highest SSIM, and the lowest JSD, indicating that the combination of the GA strategy and the SVRE loss function could help improve both forecast accuracy and spatial variability representation. The traditional optical flow method, PySTEPS, showed the least FAR and forecasting bias, while in other metrics it fell far behind our proposed model. The DL-based models (SmaAt-UNet and MotionRNN) performed worse in most metrics than the AGAN + SVRE.

*4.2. Case Study*

In this subsection, we selected two cases from the test set and further analyzed the nowcasting performances of different models for these cases.

4.2.1. Case 1

The first case was selected from a mesoscale squall line from the midsummer of 2019 on 6 August UTC. A 2-h (20-step) sequence from 15:00 to 16:54 was selected. The first half (from 15:00 to 15:54) of the sequence was fed into the model as the input, while the second half (from 16:00 to 16:54) served as the ground truth for evaluation. The +60 min radar images forecast by different models were visualized together with the observation in Figure 4. The top left subfigure indicates that the weather system evolved into a squall line trending from southwest to northwest at the lead time of 60 min. It can be easily observed that all DL models suffered from the blurry effect. The prediction of PySTEPS was closer to the observation in terms of peak reflectivity intensity, but it had significant errors in

peak positions and echo shapes. The predictions of the SmaAt-UNet and MotionRNN had opposite systematic deviations compared to those of PySTEPS. The AGAN(g) performed better than the baseline models, but it exaggerated the extent of the storm center, particularly in areas where the reflectivity was over 35 dBZ. The AGAN + SVRE provided the closest prediction to the observation in both peak intensity and center location. Compared to the AGAN(g), the blurry effect could be alleviated when the GA strategy and the SVRE loss function were simultaneously implemented.

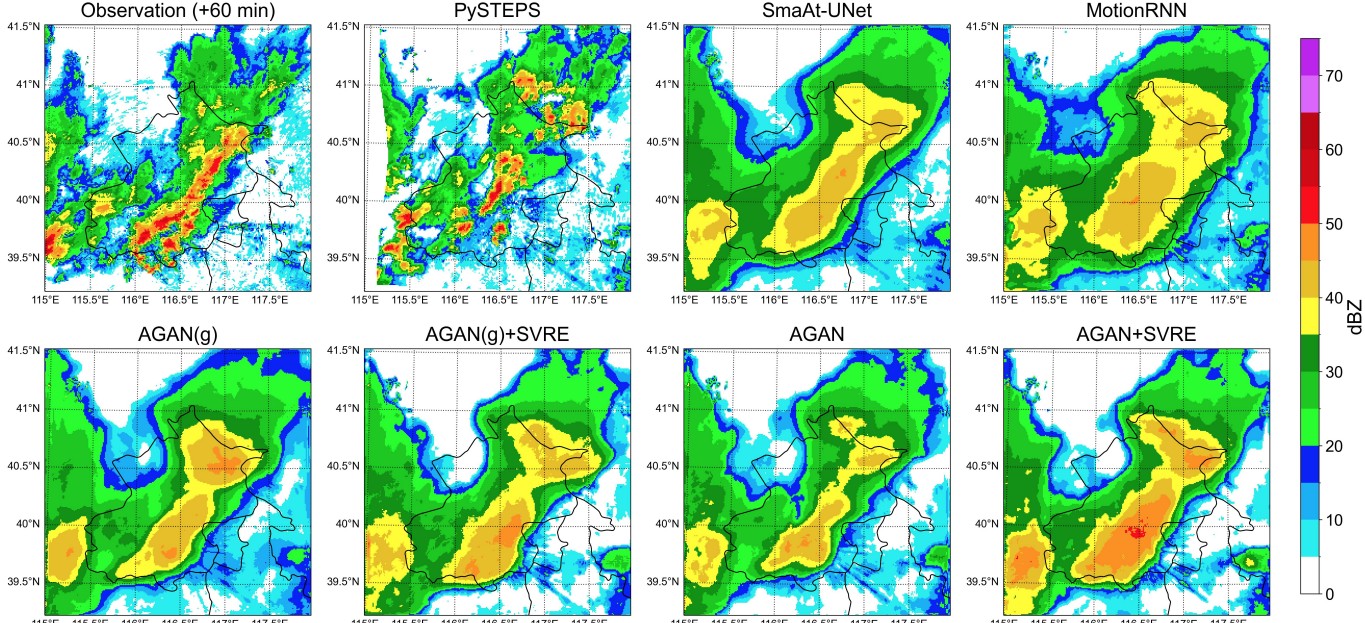

**Figure 4.** Visualization of the +60 min prediction and the observation in case 1.

A contrast scatter plot is presented in Figure 5 to evaluate the pixelwise similarity between the +60 min prediction and observation in case 1 for each model. The horizontal axis of each subfigure represents the observed value, and the vertical axis represents the predicted value. Points on the 45° line are regarded as perfect predictions, while points below or beyond this line correspond to underestimations or overestimations, respectively. The points are colored with their probability density provided by the Gaussian kernel density estimation. The results show that DL models tended to overestimate low-intensity pixels but underestimated high-intensity pixels, which coincided with the blurry effect in Figure 4. The positions of the peak probability density of the SmaAt-UNet and MotionRNN were beyond the 45° line, reflecting their systematic overestimations of mid-intensity pixels (between 20 and 30 dBZ). The AGAN(g), AGAN(g) + SVRE, AGAN, and AGAN + SVRE had lower forecast biases than the DL models. Their positions of peak probability density were more concentrated around the 45° line. PySTEPS had the most balanced prediction with the least general forecast bias, but its performance was significantly limited in pixels over 35 dBZ, which might cause a failure in extreme precipitation scenarios.

Table 4 enumerates the models' nowcasting performances for case 1. It was found that both the AGAN and AGAN(g) offered better comprehensive forecasting accuracy (higher CSI) than the baseline models. They also achieved a smaller forecasting bias (lower MBE, MAE, and RMSE) and a closer spatial variability to the observation (higher SSIM and lower JSD). When trained with the SVRE loss function, the AGAN and AGAN's generator obtained a higher SSIM and lower JSD than the SmaAt-UNet and MotionRNN, as well as a slightly reduced CSI and increased FAR. The AGAN + SVRE reached the highest SSIM (0.358) and the lowest JSD (0.616) among all the models. Although PySTEPS reached the lowest bias for the MBE, MAE, and RMSE, it was limited in regard to the POD, CSI, SSIM, and JSD because of the misplacement of the peak intensity location, mentioned above.

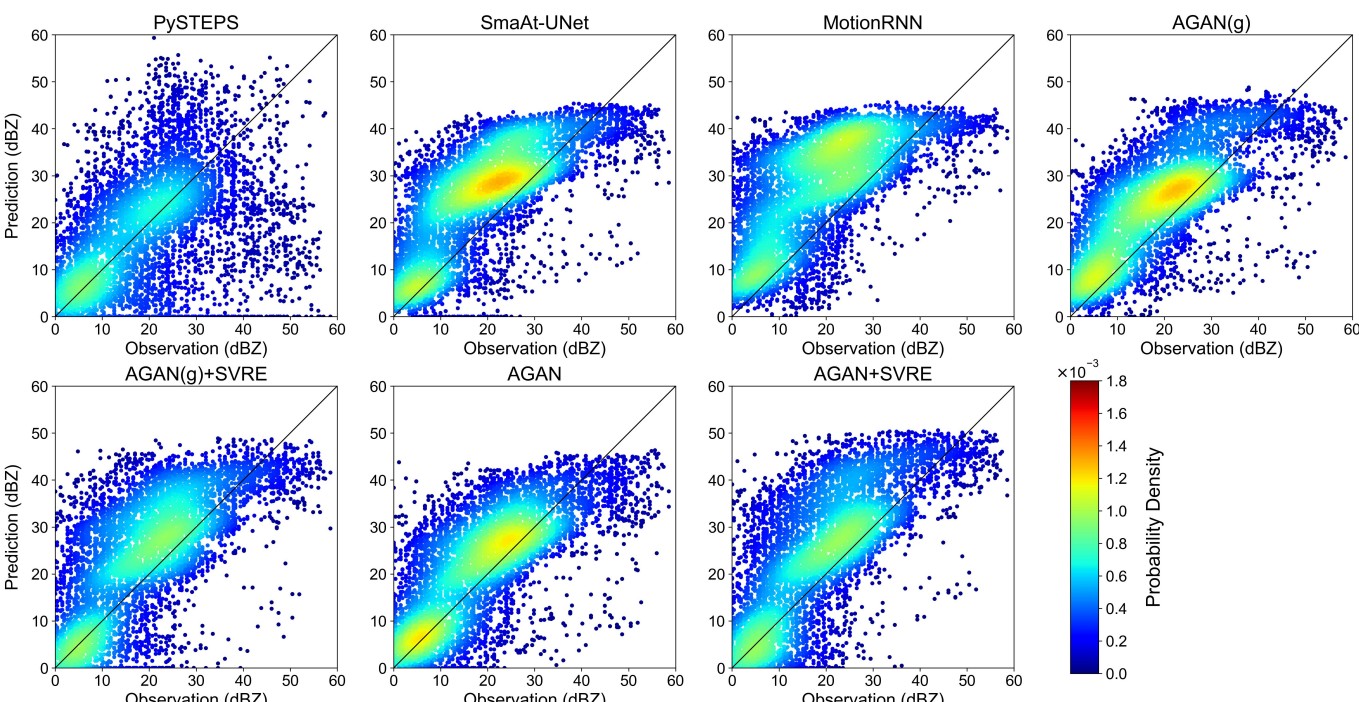

**Figure 5.** Contrast scatter plot of +60 min prediction and observation in case 1.

**Table 4.** The +60 min nowcasting performances in case 1. The up and down arrows in the heading indicate whether the highest or the lowest was the best for different metrics. A bold number indicates that the model in its row had the best performance, evaluated with the metric in its column.

| Model | POD↑ | FAR↓ | CSI↑ | MBE↓ | MAE↓ | RMSE↓ | SSIM↑ | JSD↓ |
|---|---|---|---|---|---|---|---|---|
| PySTEPS | 0.483 | **0.455** | 0.344 | −**0.2** | **8.3** | **11.8** | 0.231 | 0.672 |
| SmaAt-UNet | **0.936** | 0.556 | 0.431 | 10.4 | 11.1 | 15.1 | 0.304 | 0.664 |
| MotionRNN | 0.914 | 0.601 | 0.384 | 11.0 | 11.7 | 15.5 | 0.264 | 0.619 |
| AGAN(g) | 0.899 | 0.510 | 0.464 | 10.1 | 10.8 | 14.9 | 0.354 | 0.661 |
| AGAN(g) + SVRE | 0.881 | 0.554 | 0.421 | 9.1 | 10.5 | 14.4 | 0.317 | 0.650 |
| AGAN | 0.819 | 0.482 | **0.464** | 8.3 | 9.7 | 13.7 | 0.358 | 0.621 |
| AGAN + SVRE | 0.927 | 0.536 | 0.448 | 9.1 | 10.2 | 14.0 | **0.368** | **0.616** |

To further understand the results in Table 4, we plotted a Taylor diagram for the two experiments, which is shown in Figure 6. Taylor diagrams are widely used for the performance evaluation of meteorological models, since they can provide a concise statistical summary of the correlation coefficient (CC), the centered root mean squared error (RMSE′), and the variance ratio ($\sigma_{\hat{y}}/\sigma_y$) in a single diagram, based on the decomposition law proved by Taylor [64], which is explained in the following equation.

$$\frac{\text{RMSE}'^2}{\sigma_y^2} = \frac{\text{RMSE}^2 - \text{MBE}^2}{\sigma_y^2} = \left(\frac{\sigma_{\hat{y}}}{\sigma_y}\right)^2 + 1 - 2\frac{\sigma_{\hat{y}}}{\sigma_y} \cdot \text{CC} \qquad (17)$$

In a Taylor diagram, the overall bias of a model can be attributed to the variability part measured by $\sigma_{\hat{y}}/\sigma_y$ and the correlation part measured by CC, which are denoted by the radial axis and the circumferential axis, respectively. The overall bias can be described by the RMSE′, which is depicted by an arc with the center lying on the horizontal radial axis. From Figure 6, it can be observed that the correlation coefficients of all the DL models were concentrated around 0.7. PySTEPS had the lowest CC of around 0.6 but the highest variance ratio of over 0.9, which coincided with the visualization results. It was also observed that $\sigma_{\hat{y}}/\sigma_y$ of the AGAN + SVRE was considerably higher than $\sigma_{\hat{y}}/\sigma_y$ of the AGAN(g), implying that the introduction of the GA strategy and the SVRE loss function could narrow the spatial

variability distance between the prediction and the observation. It was also demonstrated that the AGAN + SVRE worked better in regard to the spatial variability representation than the DL-based models.

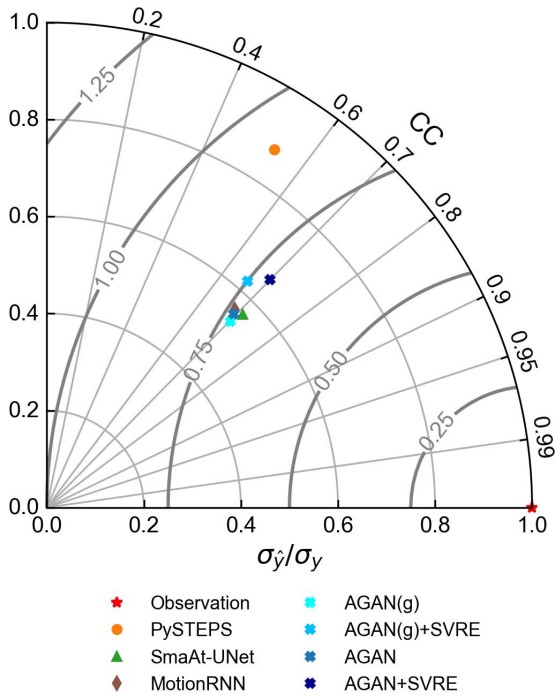

**Figure 6.** Taylor diagram of the +60 min prediction and the observation in case 1.

We also calculated the power spectral density (PSD) of both the X-axis and Y-axis of the radar maps to evaluate the models' abilities to capture local-scale weather patterns, and present the results in Figure 7. The horizontal axis of the PSD line plot was set to the logarithmic wavelength, instead of the frequency in the original concept of PSD, to more intuitively reflect the model's ability to capture weather patterns of different spatial scales [32]. If the prediction's PSD and the observation's PSD were close in a short-wavelength interval, we could say that the prediction had a similar local spatial pattern as that of the observation. The figure shows that, compared to the optical-flow-based PySTEPS method, all the DL models underestimated the PSD of the radar reflectivity along both the X-axis and Y-axis, especially for small-scale patterns (wavelength below 16 km), corresponding to the aforementioned blurry effect. It can be observed that the PSD of models trained with the GA strategy or the SVRE loss function was higher than the AGAN(g)'s PSD, indicating that the combination of the GA strategy and the SVRE loss function alleviated the systematic underestimation of spatial variability. Meanwhile, the AGAN + SVRE was ahead of the CNN-based SmaAt-UNet at all scales. The performance of the MotionRNN was the worst in small-scale local patterns with a spatial scale of fewer than 4 km, especially along the Y-axis. The results suggest that the AGAN + SVRE is more qualified for capturing local-scale patterns, which also verifies its ability to perform spatial variability representation from the side. Although the prediction of PySTEPS was the closest to the observation, the model's prediction was limited by its low correlation (CC) and forecasting accuracy (CSI).

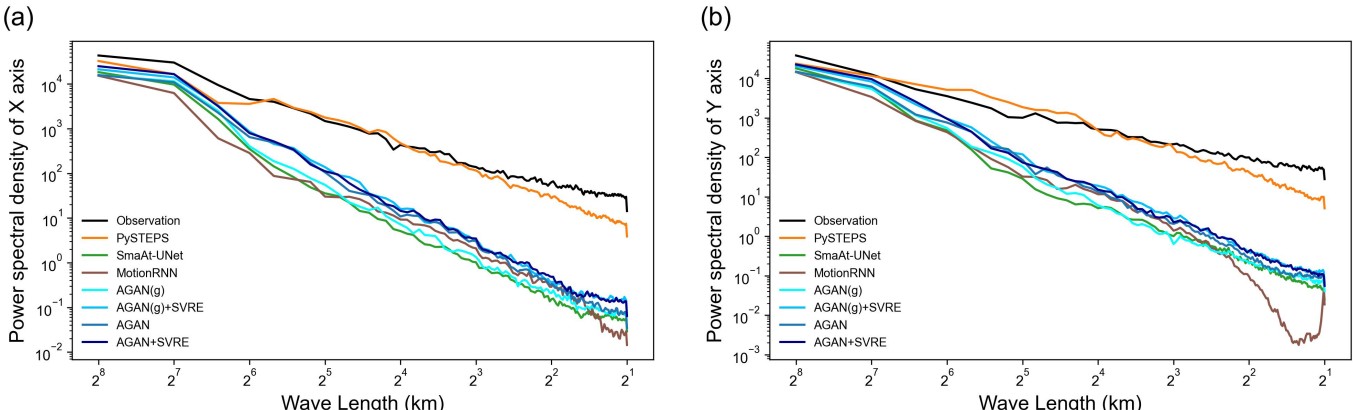

**Figure 7.** Power spectral density (PSD) of the +60 min prediction and observation in case 1. (**a**,**b**) are PSD of the X- and Y-axis, respectively.

### 4.2.2. Case 2

The second case was snipped from a growing local storm cell on 9 August 2019 UTC, lasting from 6:00 to 7:54. The +60 min predictions are visualized with the observations in Figure 8. The pixelwise nowcasting performances of case 2 were also evaluated with a contrast scatter plot, presented in Figure 9 The results show that the intensity predicted by PySTEPS at the storm center was very close to that of the observation, but the location of the storm center deviated from the observation. In contrast, DL models could successfully forecast the correct location of the storm center, whereas the underestimation of the peak reflectivity intensity and the exaggeration of the storm extent still existed, which could also be confirmed in the scatter plot. The last subfigures in Figure 8 show that the GA strategy could slightly alleviate this exaggeration effect and SVRE could increase the peak reflectivity intensity, pushing it closer to that of the observation. With the combination of the GA strategy and the SVRE loss function, the AGAN + SVRE gave the best prediction compared to the other baseline models.

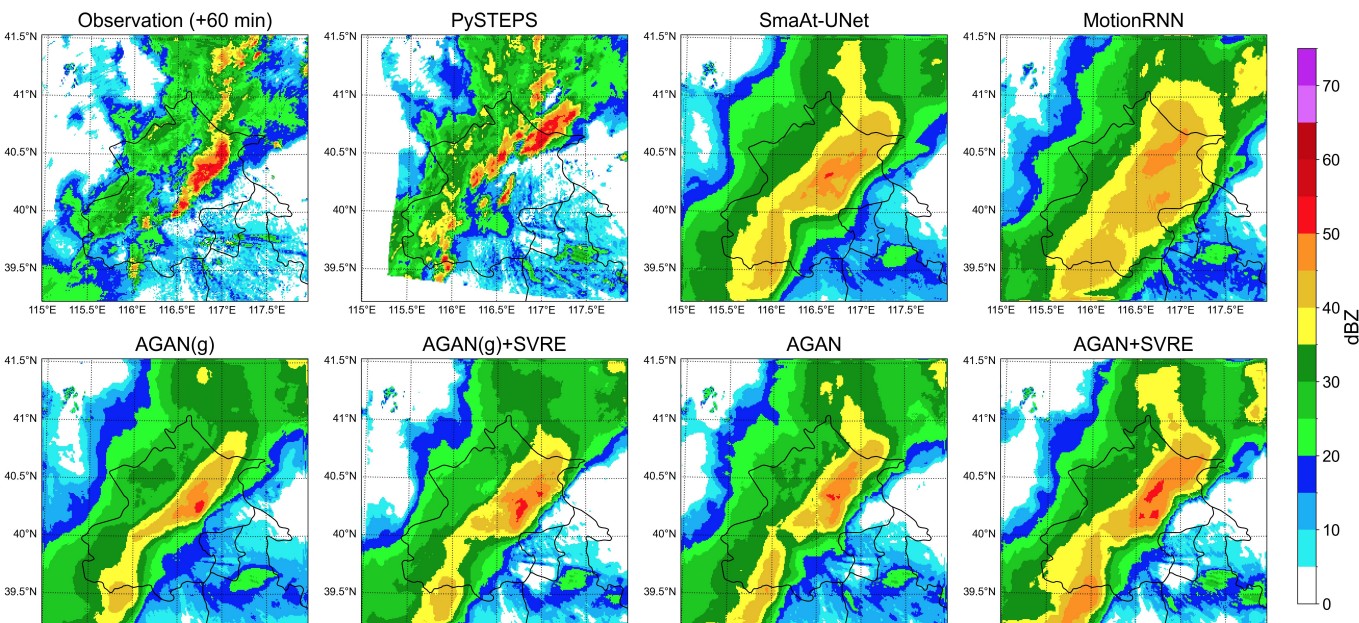

**Figure 8.** Visualization of the +60 min prediction and the observation in case 2.

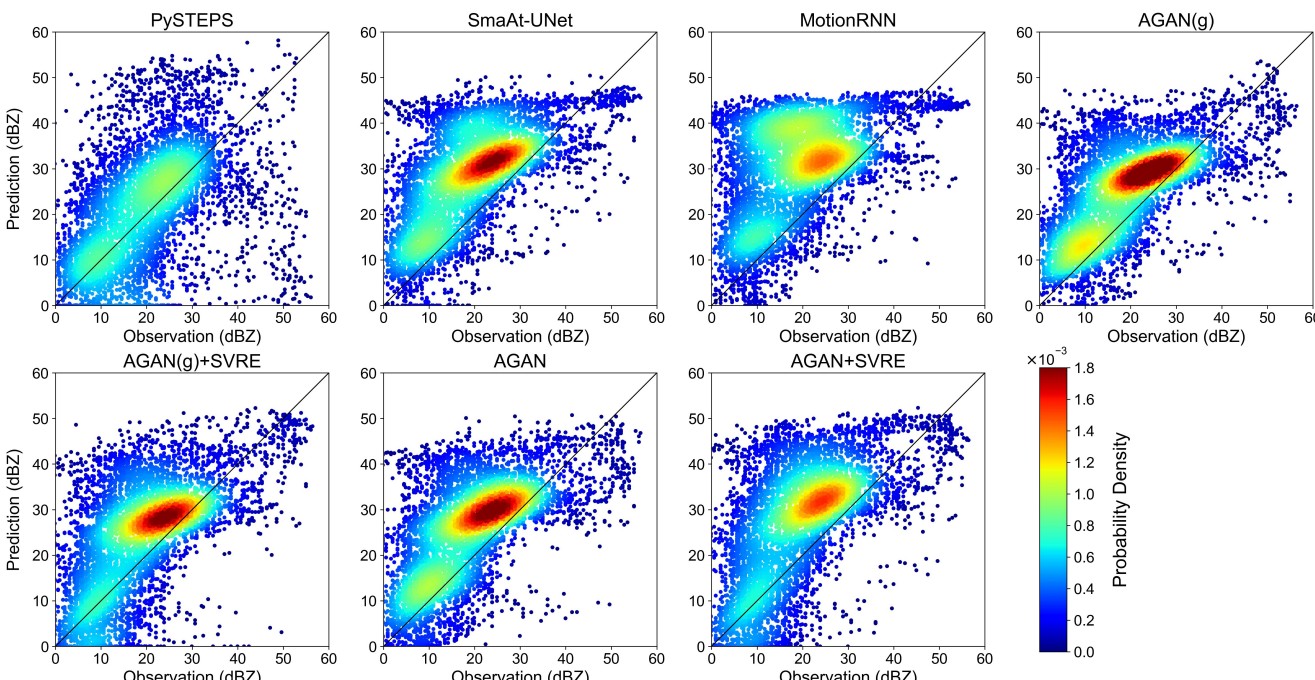

**Figure 9.** Contrast scatter plot of +60 min prediction and observation in case 2.

Table 5 enumerates the +60 min nowcasting performances of case 2. The GA strategy and the SVRE loss function improved the performance of the AGAN's generator in regard to the POD, MAE, RMSE, SSIM, and JSD, which was similar to case 1. However, the CSI of the model trained with the GA strategy and the SVRE loss function was reduced in both cases, which was different from the findings of the overall performances on the whole test set.

**Table 5.** The +60 min nowcasting performances in case 2. The up and down arrows in the heading indicate whether the highest or the lowest was the best for different metrics. A bold number indicates that the model in its row had the best performance, evaluated with the metric in its column.

| Model | POD ↑ | FAR ↓ | CSI ↑ | MBE ↓ | MAE ↓ | RMSE ↓ | SSIM ↑ | JSD ↓ |
|---|---|---|---|---|---|---|---|---|
| PySTEPS | 0.696 | **0.476** | 0.426 | **0.8** | **7.5** | **10.6** | 0.228 | 0.642 |
| SmaAt-UNet | **0.981** | 0.647 | 0.351 | 9.8 | 10.2 | 13.9 | 0.284 | 0.442 |
| MotionRNN | 0.936 | 0.659 | 0.333 | 10.7 | 11.3 | 15.3 | 0.249 | 0.298 |
| AGAN(g) | 0.933 | 0.519 | **0.465** | 8.3 | 8.8 | 13.2 | 0.306 | 0.331 |
| AGAN(g) + SVRE | 0.873 | 0.524 | 0.445 | 7.0 | 8.2 | 12.0 | 0.305 | 0.409 |
| AGAN | 0.951 | 0.604 | 0.388 | 8.5 | 9.1 | 12.8 | 0.327 | 0.341 |
| AGAN + SVRE | 0.960 | 0.617 | 0.378 | 8.3 | 9.1 | 12.8 | **0.332** | **0.271** |

We also plotted the Taylor diagram (Figure 10) and the PSD line chart (Figure 11) for case 2. In the Taylor diagram, it was found that both the GA strategy and the SVRE loss function could promote $\sigma_{\hat{y}}/\sigma_y$ of the model, and SVRE had an even bigger promotional effect than GA. The RMSE', related to the overall bias, was consequently reduced. For the comparison experiment, the advantage of AGAN + SVRE over other baseline models was similar to case 1. It achieved the highest CC, the lowest RMSE', and the second-highest variance ratio below PySTEPS. For the PSD of the predictions in Figure 11), the similarities and differences of the nowcasting performances between the models in case 2 were homogeneous to those in case 1. The PSD of radar images predicted by the AGAN + SVRE were the closest to that of the observation among all of the DL models.

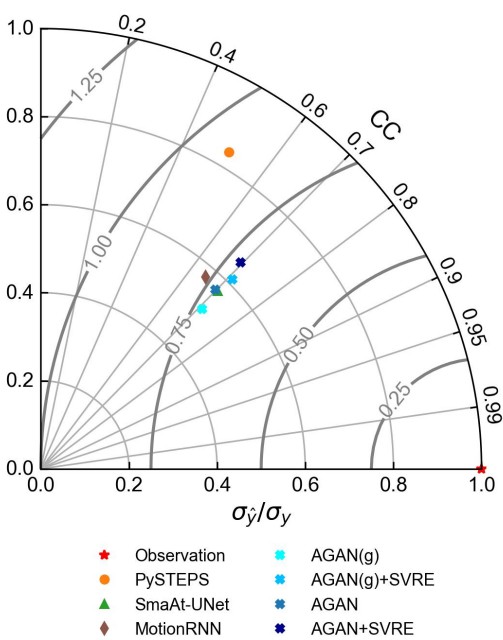

**Figure 10.** Taylor diagram of the +60 min prediction and the observation in case 2.

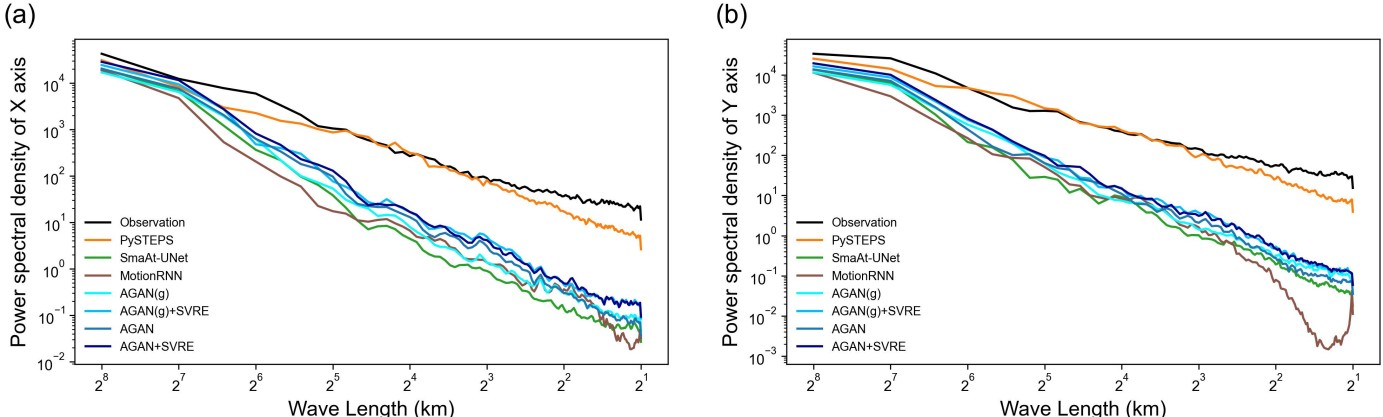

**Figure 11.** Power spectral density (PSD) of the +60 min prediction and observation in case 2. (**a**,**b**) are PSD of the X- and Y-axis, respectively.

### 4.3. Discussion

Thus far, we have evaluated the overall performances of different models and further analyzed their strengths and shortcomings with two storm cases. Generally, the results proved that the GA strategy and the SVRE loss function could alleviate the blurry effect of DL nowcasting models. The SVRE loss function and the GA strategy could boost DL nowcasting models by bridging the gaps between the predictions and the observations, particularly regarding spatial variability differences. As expected, the spatial variability-related metrics (SSIM, JSD, and PSD) of the test set and the two selected cases demonstrated that the GA strategy and the SVRE loss function could enhance the spatial variability representation of DL nowcasting models. More specifically, the enhancement of spatial variability representation derived from SVRE instead of GA. This might have been influenced by the hyperparameter selection in generative adversarial training. Since GANs are difficult to train, a larger weight has to be attached to the SVRE loss term and the reconstruction term to ensure convergence, leading to enhanced difference between the two components.

However, our methods also had several limitations. The first was that the forecasting accuracy of models trained with the SVRE loss function slightly reduced in heavy-rainfall



cases. The AGAN(g)+SVRE and AGAN + SVRE could improve the CSI of the AGAN(g) and AGAN on the whole test set, but they failed in the two storm cases. This was probably because the goal of better spatial variability representation of extreme storm events can lead to more overestimated pixels or false alarms, which have a negative impact on forecasting accuracy. Another limitation came from the computational cost. The generative adversarial training process and the approximated integration over latent variables can significantly increase the convergence time for the training and inference processes of our model, making it difficult to apply to scenarios that require an extremely rapid response.

## 5. Conclusions

Previous deep learning models for radar nowcasting suffer from the systematic "blurry" problem and do not accurately represent the spatial variability of radar echo images. This study presented a Spatial Variability Representation Enhancement (SVRE) loss function and an Attentional Generative Adversarial Network (AGAN) to solve the problem, and evaluated them with a regional CINRAD dataset. An ablation experiment and a comparison experiment were implemented to verify the effects of the generative adversarial (GA) training strategy and SVRE loss and to compare the proposed model to current advanced radar nowcasting models. The performances of the models were validated on the whole test set and then inspected in two typical cases. Several metrics were selected to evaluate the forecasting accuracy and spatial variability representation. The results showed that both the GA strategy and the SVRE loss function could improve nowcasting performance by enhancing the spatial variability representation of the radar reflectivity. The GA strategy and SVRE also helped our model outperform other advanced baseline nowcasting models. The main contributions of this study are the following:

- We propose the SVRE loss function and the AGAN to alleviate the blurry effect of DL nowcasting models. Both of them can reduce this effect by enhancing the spatial variability of radar reflectivity.
- We attribute the blurry effect of DL nowcasting models to the deficiency in spatial variability representation of radar reflectivity or the precipitation field, which provides a new perspective for improving radar nowcasting.

Consequently, this study provides a feasible solution based on dense radar observations for high-resolution radar nowcasting applications. The limitations of our methods include reduced forecasting accuracy in high-intensity storm events and heavy computation costs. Future studies will focus on overcoming these limitations for high-intensity and small-scale storm events.

**Author Contributions:** Conceptualization, A.G.; methodology, A.G.; software, A.G.; validation, A.G.; formal analysis, A.G.; investigation, A.G. and R.L.; resources, B.P. and H.C.; data curation, M.C.; writing—original draft preparation, A.G. and R.L.; writing—review and editing, H.C., B.P. and G.N.; visualization, A.G. and R.L.; supervision, G.N.; project administration, G.N.; funding acquisition, G.N. All authors have read and agreed to the published version of the manuscript.

**Funding:** This work was supported by the National Key Research and Development Program of China (2018YFA0606002) and the Fund Program of State Key Laboratory of Hydroscience and Engineering (61010101221). The work of Haonan Chen was supported by Colorado State University.

**Data Availability Statement:** The codes for the experiments and the results can be found at https://github.com/THUGAF/SVRE-Nowcasting (accessed on 14 June 2023). The intermediate products, such as the feature maps and the pre-trained models, are available upon request.

**Acknowledgments:** The authors gratefully acknowledge the anonymous reviewers for providing careful reviews and comments on this article.

**Conflicts of Interest:** The authors declare no conflict of interest.

## Abbreviations

The following abbreviations are used in this manuscript:

| | |
|---|---|
| SVRE | Spatial varibility representation enhancement |
| AGAN | Attentional generative adversarial network |
| DL | Deep learning |
| GA | Generative adversarial |
| CNN | Convolutional neural network |
| RNN | Recurrent neural network |
| POD | Probability of detection |
| FAR | False alarm ratio |
| CSI | Critical sucess index |
| MBE | Mean bias error |
| MAE | Mean absolute error |
| RMSE | Root mean squared error |
| JSD | Jensen-Shannon divergence |
| SSIM | Structural similarity index measure |
| PSD | Power spectral density |
| CC | Correlation coefficient |

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
