# Peer review of "Enhancing Spatial Variability Representation of Radar Nowcasting with Generative Adversarial Networks"

_remotesensing, doi:10.3390/rs15133306_

Round 1

Reviewer 1 Report

In this article, a new weather nowcasting model based on the radar spatial variability representation, correcting the blurry effects derived from deep learning methods, is presented.

To validate the algorithm, results derived from the radar network CINRAD were presented.

This paper appears to make a good contribution to the literature. However, the absence of addressed bibliographic references makes it difficult to follow the document.

Apart from this, my comments mostly focus on better highlighting the contribution of the study:

·         The main objective of this work is to correct the blurry effects presented in the deep learning methods. Where can this effect be seen? or Where is this effect corrected? I think that the improvements in this point are not shown properly.

·         In the Power Spectral Density figures, what does the Wave Length axis actually represent? It is not sufficiently clear if it is related to the radar frequency.

Reviewer 2 Report

Title: Enhancing Spatial Variability Representation of Radar Nowcasting

Authors: Aofan Gong, Ruidong Li, Baoxiang Pan, Haonan Chen, Guangheng Ni and Mingxuan Chen

Recommendation: Accept

Summary: This study presents a Spatial Variability Representation Enhancement (SVRE) loss function and proposes an effective nowcasting model named Attentional Generative Adversarial Network (AGAN) to alleviate this blurry effect by enhancing the spatial variability representation in radar nowcasting tasks. Results show that both GA strategy and SVRE loss can enhance the spatial variability representation and narrow the forecasting bias, which helps AGAN achieve better nowcasting performance than the other competitor models. However, there are some aspects missing in the manuscript that need to be addressed, and I propose to publish this paper after revision.

Minor Comments:

1)  Please check the references marks throughout the manuscript.

2)  Figure2(b): Should the output feature dimension here be Cout×H×W?

3)  Figure6(d): It is recommended to align the explanation position of the icon with (a)(b)(c). Alternatively, it can be placed uniformly in the bottom left corner.

4)  Figure6(d) and 9(d): The curve of MotionRNN shows a rapid decrease and then an increase in the small wavelength region. You mention "unacceptable", can you explain this phenomenon in detail.

5)  Table 5: I understand that bold is the most effective indicator, but in the CSI indicator, you have highlighted two items in bold. Please explain the reason.

6)  In line 159: The font of "least-squares loss" is incorrect.

7)  In line 277:  and  have been explained in line 257, here it should be  and .

8)  A brief explanation of what the numbers in bold in the table represent should be given.

9)  Line 293: How is the learning rate determined for generator and discriminator?

10)  Line 407: The word "deviated" is used incorrectly and is not normally used in the passive tense.

11)  In section 5.2, the results in Tables 4 and 5 lack analysis of the results for longitudinal comparisons and explanations of the reasons.

12)  In section 5.2. Case Study, please add the contrast scatter plot to show the similarity between the model prediction plot and the real observation plot more intuitively.

Reviewer 3 Report

The study “Enhancing Spatial Variability Representation of Radar Nowcasting” presents an interesting and important study for the scientific community. However, before recommending the present study for publication, several points need to be corrected. Therefore, I am considering this manuscript for major revisions, highlighting as main observations:

1 – The authors did not respect the formatting of the manuscript in the journal's template, a complete adaptation to the journal's template is required!

2 – The manuscript does not present bibliographical references and much less citations, the authors must make this correction!

3 – Structure the manuscript in: 1. Introduction; 2. Material and Methods; 3. Results and Discussion; 4. Conclusions; 5. References.

In its current state, it is quite difficult to interpret the logic of the research as a whole.

4 – Standardize the dimensions of Figures 1 and 2!

After these changes, I would like to receive the manuscript again for a final reading.

Even though I am not fluent in the English language, I noticed several grammatical and spelling errors, so I suggest a review by a native English speaker.

Reviewer 4 Report

radar echo extrapolation 4 methods

Weather radars image is very important to weather nowcasting tasks, but it is challenged by radar echo extrapolation methods to reduce spatial “blurry” effect, the manuscript mainly proposed a   method to enhance the spatial variability representation in radar nowcasting tasks, the results sounds better and meaningful.

But there are some key issues to be addressed for the authors as follows:

1) the title should be revised to clear the research contents.

2) The cited reference is ? all through the manuscript.

3)about Figure 3.  Why did you use Topographic background map? we defifined the study area as a square region of 256×256 km2” ,could you give the legend to show the square region?

4) what did you mean about the ablation experiment ? could you add information to describe it or use more clear word instead of ablation?

5) In Table 2.  From POD, we can see that AGAN(g) is the best, form FAR, AGAN+SVRE is the wowhat did you mean aboutrst, how to explain the results?

6)what did you mean about the lead time of 60 min”.  could you add information to describe or use more clear word instead of lead?

7)Indeed, the performance of radar echo extrapolation methods is very related to the Predictive time length, the research is only give a research of 60 min prediction, could you add the comparison the performance with time variation?

8) In Table 5, From  POD, AGAN(g)+SVRE is getting worse than AGAN(g), from FAR, AGAN+SVRE is also  is getting worse than AGAN, From  POD, SmaAt-UNet is the highest, which mean SVRE  showed not more better performance, could you plz explain it?

9)In Figure 6 and Figure 9, from C and D,  the optical-flflow-based PySTEPS method it the most close to the observation, but the method added SVRE looks like a littlle improved, so the result should be more explained and discussed to show the performance of your proposed methods.

10) the writing had better be improved to be more clear and understood well for readers. 

Round 2

Reviewer 1 Report

I congratulate the authors to provide a detailed rebuttal to the major comments. Then, the paper is OK to be published.

Reviewer 3 Report

The requested corrections have been made. Therefore, I am accepting this study for publication.

The requested corrections have been made. Therefore, I am accepting this study for publication.